# Should We Quantify Valvular Calcifications on Cardiac CT in Patients with Infective Endocarditis?

**DOI:** 10.3390/jcm10194458

**Published:** 2021-09-28

**Authors:** Virgile Chevance, Remi Valter, Mohamed Refaat Nouri, Islem Sifaoui, Amina Moussafeur, Raphael Lepeule, Eric Bergoend, Sebastien Mule, Vania Tacher, Raphaelle Huguet, Thierry Folliguet, Florence Canoui-Poitrine, Pascal Lim, Jean-François Deux

**Affiliations:** 1AP-HP, Hopital Henri Mondor, Service de Radiologie, IMRB, F-94010 Créteil, France; mohamedrefaat.nouri@aphp.fr (M.R.N.); islem.sifaoui@aphp.fr (I.S.); sebastien.mule@aphp.fr (S.M.); vania.tacher@aphp.fr (V.T.); deuxjf@gmail.com (J.-F.D.); 2AP-HP, Hopital Henri Mondor, Service de Santé Publique, IMRB, F-94010 Créteil, France; remi.valter@aphp.fr (R.V.); florence.canoui-poitrine@aphp.fr (F.C.-P.); 3AP-HP, Hopital Henri Mondor, Service de Cardiologie, IMRB, F-94010 Créteil, France; amina.moussafeur@yahoo.fr (A.M.); raphaelle.huguet@aphp.fr (R.H.); pascal.lim@aphp.fr (P.L.); 4AP-HP, Hopital Henri Mondor, Département de Prévention, Diagnostic et Traitement des Infections, IMRB, F-94010 Créteil, France; raphael.lepeule@aphp.fr; 5Service de Chirurgie Cardiaque Assistance Publique-Hôpitaux de Paris (AP-HP), Hôpitaux Universitaires Henri Mondor, F-94010 Créteil, France; eric.bergoend@aphp.fr (E.B.); thierry.folliguet@aphp.fr (T.F.); 6Unité INSERM U955 Team 18, IMRB, F-94010 Creteil, France; 7Clinical Epidemiology and Ageing Unit, Institute Mondor de Recherche Biomédicale, Paris-Est University, F-94000 Créteil, France

**Keywords:** cardiac computed tomography, endocarditis bacterial, valve calcium scoring, cardiac imaging techniques

## Abstract

Background: Evaluate the impact of valvular calcifications measured on cardiac computed tomography (CCT) in patients with infective endocarditis (IE). Methods: Seventy patients with native IE (36 aortic IE, 31 mitral IE, 3 bivalvular IE) were included and explored with CCT between January 2016 and April 2018. Mitral and aortic valvular calcium score (VCS) were measured on unenhanced calcium scoring images, and correlated with clinical, surgical data, and 1-year death rate. Results: VCS of patients with mitral IE and no peripheral embolism was higher than those with peripheral embolism (868 (25–1725) vs. 6 (0–95), *p* < 0.05). Patients with high calcified mitral IE (mitral VCS > 100; *n* = 15) had a lower rate of surgery (40.0% vs.78.9%; *p* = 0.03) and a higher 1-year-death risk (53.3% vs. 10.5%, *p* = 0.04; OR = 8.5 (2.75–16.40) than patients with low mitral VCS (*n* = 19). Patients with aortic IE and high aortic calcifications (aortic VCS > 100; *n* = 18) present more frequently atypical bacteria on blood cultures (33.3% vs. 4.8%; *p* = 0.03) than patients with low aortic VCS (*n* = 21). Conclusion: The amount of valvular calcifications on CT was associated with embolism risk, rate of surgery and 1-year risk of death in patients with mitral IE, and germ’s type in aortic IE raising the question of their systematic quantification in native IE.

## 1. Introduction

Cardiac computed tomography (CT) has the capability to detect, localize and quantify the calcifications in coronary arteries [1] and cardiac valves [2,3,4,5,6], information that may be of interest for the management of patient with cardiovascular diseases. Quantitative measurement of coronary calcifications (also called coronary calcium scoring) is highly correlated with the occurrence of major cardiovascular disease [1,7] and is widely used to improve the assessment of the cardiovascular risk of patients with intermediate probability of cardiovascular diseases [8,9,10]. CT quantification of aortic valve calcifications (valvular calcium score-VCS) is also commonly employed to assess the degree of severity of aortic stenosis [3,4,5,11,12,13]. Aortic VCS is a strong predictor of severity of aortic stenosis in patients with low flow-low gradient reduced ejection fraction and more widely in case of aortic stenosis difficult to evaluate with transthoracic echocardiography (TTE) [4]. Evaluation of valve and perivalvular calcifications before transaortic valve implantation (TAVI) [14,15,16,17] or transcatheter mitral valve replacement (TMVR) are also of interest [6]. Indeed, marked annular or leaflet aortic calcifications may be displaced by prosthesis and induced ischemic or rhythmic complications [16,18,19]. Lastly, in case of surgical valve treatment, the degree of valve calcifications (and degeneration) is an important data regarding the possibility to perform a valve repair or a replacement surgery [20,21,22]. Infective endocarditis (IE) is a severe disease with a high mortality rate of up to 40% [23,24]. The diagnosis of IE is established using the modified Duke ESC criteria based on biological and imaging findings [25,26]. Transesophageal echocardiography (TEE) is considered as the technique of choice to diagnose native and prosthetic IE and to detect valvular and paravalvular IE-related cardiac complications, such as vegetation, valvular perforation, new dehiscence of prosthetic valve, abscess and pseudoaneurysm [26]. Since several years, cardiac CT has emerged as a first line technique to analyse IE-related cardiac lesions in addition to echocardiography [27,28,29,30,31,32,33,34,35,36,37,38]. The presence of paravalvular lesions on cardiac CT images was proposed recently as a major criteria for the diagnosis of IE on the modified ESC Dukes criteria [26,39,40]. To the best of our knowledge, none study has evaluated the interest to detect and quantify valve calcifications on CT images in patients with left-sided IE. We hypothesized that the quantification of mitral and aortic valve calcifications on CT images could be a relevant data to add in CT reports in patients with IE, and could provide relevant insights regarding patient status, management and prognostic. In this study, we aim to evaluate the relationships between valve calcifications quantification with cardiac CT and clinical, biological, surgical data as well as 1-year death rate, in a large cohort of patients with IE.

## 2. Materials and Methods

### 2.1. Population

All patients referred to our institution with a definite diagnosis of native left-sided IE on modified Dukes criteria [25,26] and a cardiac-CT in our institution were retrospectively included in this study between January 2016 and April 2018 (*n* = 143). Exclusion criteria were patients with right heart infective endocarditis (*n* = 4), patients explored with non-gated cardiac CT (*n* = 12), prosthetic valve IE (*n* = 51) and lack of unenhanced images in CT protocol (*n* = 6). Finally, 70 patients were included (Figure 1).

In accordance with the Helsinki Declaration, the retrospective nature of the data analysis and the absence of changes to the usual care protocol (this is non-interventional research), we did not make a specific request to the patient protection committee (French Jardé law, 2017, category 3). However, all patients with an examination in our institution are invited to read and sign a document at the time of their examination authorizing the department to anonymously analyze this data for medical research purposes.

### 2.2. Acquisition Protocol

All patients were examined using a 256-row detector CT system (Revolution CT, General Electric Healthcare, Milwaukee, WI, USA). A gated calcium scoring acquisition was used for all patients with the following parameters: gantry rotation time 280 ms; tube current 120 kV; 350 mAs; FOV 250 mm; slice thickness 1.5 mm. Beta-blockers were not used in this emergency indication. Dose–length product (DLP) was recorded for all patients, and effective radiation dose calculated.

### 2.3. Post Processing

Images were analysed in consensus in blinded matter of the results of the TEE on dedicated platform (ADW server version 4.6, GE, Milwaukee, WI, USA) by 2 readers (V.C. and J.F.D) with 7 and 17 years of experience in cardiac imaging, respectively. Measurement were performed using a dedicated software (Smartscore; General Electric Healthcare, Milwaukee, WI, USA) implementing the Agatston method [1]. Lesion-specific scores were calculated as the product of the area of each calcified focus and peak CT Hounsfield unit value. A manual contouring of both mitral and aortic calcifications (including annular and leaflet calcifications) was performed on axial slices. Multiplanar reconstructions (MPR) were used to facilitate selection of valve calcifications on axial slices (Figure 2 and Figure 3). Mitral and aortic VCS were recorded, and binarized in 2 classes in function of amount of calcifications: high value of VCS (>100) and low value of VCS (≤100).

### 2.4. Patient’s Clinical Data

Clinical data’s, microbiological data’s (i.e., identification of *Streptococcus* sp, *Staphylococcus* sp. and non-staphylococcus and non-streptococcus germs called “other bacteria” on blood cultures or on culture of cardiac valvular materials obtained at surgery), presence of peripheral embolism (cerebral, spleen, kidneys, peripheral arteries, renal arteries and digestive arteries), the size of the mean vegetation size on TEE, the rate of surgery, the type of valve surgery (valve repair or replacement) and all-cause of death during the first year after admission were recorded for all patients.

### 2.5. Statistical Analysis

Categorical variables were described as numbers (percentages) and continuous variables as mean ± standard deviation (SD) or median (interquartile range) according to their distribution. Differences between groups (VCS ≤ 100 vs. VCS > 100) were determined by the Wilcoxon rank sum test for continuous variables, and χ^2^ test or Fisher’s exact test for categorical variables. VCS was also handled in continuous way. The association of VCS with the type of bacteria was determined using a multinomial log-linear model. Odds-Ratio (OR) and their 95 % confidence interval (CI) were calculated for 1-year risk of death after the diagnosis of IE. All tests were two-sided and a *p* value < 0.05 was considered significant. All statistical analyses were performed using the R Epi package [41], the R-net package [42], R logistf package [43] and R software v. 3.6.1 (R Core Team, 2019) [44].

## 3. Results

### 3.1. Population Characteristics

Characteristics of the 70 patients are summarized in Table 1. The median age was 75 (61.5–83.5) year-old and 40% of patients were female. Streptococcus was the most common organism (48.5% of cases) followed by Staphylococcus (35.7%), and other bacteria (15.7%). IE involved 36 aortic valves, 31 mitral valves, and 3 IE were bivalvular. Twenty-nine patients (41.4%) had a peripheral embolism. Mean vegetation size on TTE was 13 ± 5 mm. Fifty patients (71.4%) underwent cardiac surgery, 36 of them (72.0%) had a valve replacement (27 aortic and 9 mitral bioprosthesis) and the remaining 23 patients (46.0%) had a valve repair (4 aortic and 19 mitral valves). Patients who underwent cardiac surgery were significantly younger than non-operated patients (65 ± 16 vs. 77 ± 15 years old, *p* < 0.05). Fourteen patients (20%) died within the first year after the diagnosis of IE. Of these patients, 4 had cardiac surgery.

### 3.2. Global VCS Measurement

In the overall population, median mitral VCS was 22 (0–1010) and median aortic VCS was 16(0–374). They were significantly correlated (*r* = 0.51, *p* < 0.05), and correlated with age of patients (*r* = 0.42, *p* < 0.001 both for aortic and mitral valves).

#### 3.2.1. Patients with Native Mitral IE

Thirty-four patients with native mitral IE (31 mitral IE and 3 bivalvular IE) were analyzed (Table 2). Median value of mitral VCS was 12 (0–886). Fifteen patients (44.1%) exhibited high mitral calcifications (defined as a mitral VCS > 100; median value 2119 (797–3046)) and the remaining 19 patients (55.8%) had low mitral calcifications on CT (VCS < 100; median value 0 (0–6)). High calcified patients were significantly older (80.5 vs. 58.0 years; *p* < 0.001) than low calcified ones. Type of germ and mean vegetation size were similar between the 2 groups of patients. Patients with high amount of valvular calcifications trended to present less frequently peripheral embolism in comparison to low calcified ones but the difference did not reach statistical significance (20.0% vs. 47.4%; *p* = 0.08). However, this difference was statistically significant when comparing mitral VCS: patients without peripheral embolism had a significantly higher VCS than those with peripheral embolism (868 (25–1725) vs. 6 (0–95), *p* < 0.05).

High calcified patients exhibited a significant lower rate of surgery (40.0%) than low calcified ones (78.9%); *p* = 0.03). Similarly, when regarding continuous values of VCS, median mitral VCS of non-operated patients was significantly higher than operated one (473 (75–913) vs. (0 (0–102); *p* < 0.05). The percentage of valve replacement and surgical plasty were similar between high and low calcified patients. High calcified patients (VCS > 100) exhibited a significantly higher risk of death during the first year after the diagnosis of IE (53.3% vs. 10.5%, *p* < 0.05; OR = 8.5 (2.75–16.40) than low calcified ones. All data are reported in Table 2.

#### 3.2.2. Patients with Native Aortic IE

Thirty-nine patients with native aortic IE (36 aortic IE and 3 bivalvular IE) were analyzed (Table 3). Median value of aortic VCS score was 13 (0–423). Eighteen patients (46.1%) had high valvular aortic calcifications (VCM > 100; median value: 623 (237–1330)) and the remaining 21 patients (53.9%) had low calcifications on CT (VCM < 100; median value 0 (0–8)). Of note, 4 patients (10%) had a VCS > 2000 suggesting severe aortic stenosis.

High calcified patients (VCS > 100) were significantly older than low calcified (VCS < 100) ones (76.5 vs. 54.0; *p* < 0.001). They present more frequently non-staphylococcus and non-streptococcus bacteria on blood cultures or on culture of removed valve fragments (33.3% vs. 4.8%, *p* = 0.03) (Figure 4). The mean size of vegetation, the number of peripheral embolism, the percentage and the type of surgery, and 1-year death rate were similar between high and low calcified patients. Mean value of aortic VCS was also similar between operated (21 (0.0–228) and non-operated (38 (0–574) patients (*p* = 0.27). All data are reported in Table 3.

Frequency of detection of staphylococcus, streptococcus and atypical germs on blood cultures (and/or removed valve fragments) in patients with aortic IE, in function of aortic valvular calcium score. Gray columns represent high calcified patients (VCS > 100) and white columns represent low calcified ones. Atypical germ (i.e., non staphylococcus non streptococcus) are more frequently encountered in high calcified patients (*p* = 0.03).

### 3.3. Radiation Exposure

Mean DLP was 71.8 ± 35.7 mGy.cm for patients. The effective radiation dose was 1.22 ± 0.61 mSv. There was no significant difference for DLP between low and high calcified patients, both for aortic (69.2 vs. 73.6 mGy·cm; *p* = 0.52) and mitral (68.0 vs. 74.2 mGy·cm; *p* = 0.47) IE.

## 4. Discussion

In this study, we report a statistical association for patients with a native mitral IE and presenting high mitral calcifications 1-with a lower risk of peripheral embolism, 2-with a less frequently surgery, and 3-with a higher 1-year death rate after initial diagnosis, in comparison with low calcified patients. In aortic IE, high-calcified patients present more frequently atypical bacteria on blood cultures or on cardiac valvular material, while valve calcifications did not seem to influence surgical treatment and survival. Our data suggested that a quantitative evaluation of valve calcifications on CT may be considered in the assessment of native IE, especially in mitral IE.

Cardiac CT may improve detection of paravalvular complications in IE in addition to TEE [26,39,40], but the presence and the amount of valve calcifications are usually not taken account in images analysis. To the best of our knowledge we report here the first study that addressed the problem of detection and quantification of valve calcifications in IE with CT. In the light of our results, measurement of VCS may be of interest in mitral IE because of its impact on surgery and 1-year survival. In order to explain these results, we hypothesis that mitral valvular calcifications may be an indirect marker of the general condition of the patient and comorbidities. Mitral VCS may therefore be used as an additional prognosis factor, relatively easy to provide on CT images. We also noticed a lower frequency of peripheral embolism in high calcified mitral IE in our study (vs. low calcified patients), that seems more difficult to explain. We hypothesis that the amount of valve calcifications may modify the degree of adhesion of vegetations on valve structures and may therefore influence the risk of peripheral embolism. Future studies with larger numbers of patients and possibly prospective to allow adjustment for confounding factors are useful to explore these exploratory results.

The value of measuring the VCS in aortic IE seems less obvious, regarding our results. We did not detect a significant relationship between amount of aortic valve calcifications measured on CT and presence of embolism, type of treatment and survival. However, interestingly, we reported that atypical bacteria (i.e., non-staphylococcus and non-streptococcus bacteria) were more frequently encountered in high calcified patients. These results could suggest that the bacterial affinity for the valve tissue may be influenced by the valve calcifications. Further studies are required to valid this hypothesis and to understand the microbiological mechanisms that support these differences.

Some limits have to be mentioned in this study. First of all, this is a single-center report including a relatively low number of patients. Second, histological quantification of valve calcifications was not carried after surgery on the removed valves, precluding a correlation with VCS measured on CT images. Third, annular and leaflet calcifications were quantified together. We have not made isolated measurements of the calcifications of the leaflets, which are often more affected than the ring in endocarditis. Lastly, measurement of VCS was not feasible in case of prosthesis IE, a fairly common clinical situation.

Regarding the future perspectives for the use of the valvular calcium score in IE, we point out that this quantitative parameter is relatively easy to provide on CT images and could be of an additional parameter in future studies on patients with IE. The measurement of the calcium score series takes only a few minutes, does not require contrast medium injection and does not induce any additional cost. The post-treatment method is simple and can be perform in a few minutes. In addition, it is already widely used in other indications such as before TAVI implantation. Its real clinical utility in IE remains to be confirmed, but we present some insights for its potential interest in embolic risks and survival of patients. The relationship of VCS with the type of germ is also original and could bring new hypothesis in the development of the germs in IE.

## 5. Conclusions

Patients with highly calcified mitral IE had a lower risk of peripheral embolism, had less frequently surgery, and experienced a lower 1-year survival after initial diagnosis in comparison with low calcified patients. Patients with highly calcified aortic IE present more frequently atypical germs than low calcified one. Further studies with a large number of patients are required to confirm these results.

## Figures and Tables

**Figure 1 jcm-10-04458-f001:**
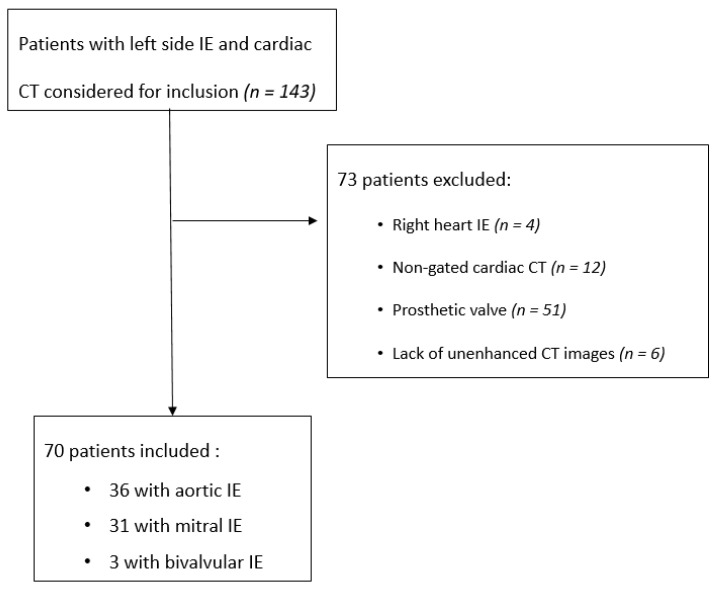
Flowchart of the patients.

**Figure 2 jcm-10-04458-f002:**
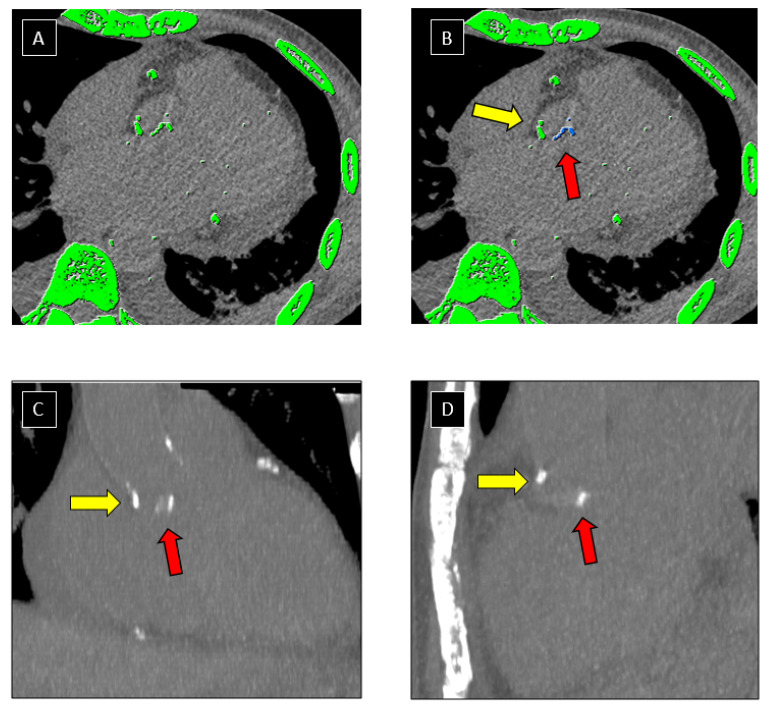
Post processing of CCT images. Example of segmentation of aortic valve calcifications on CT images using a dedicated software. Automatic detection of structures with a density > 130 Hounsfield Units (highlighted in green) was performed by the software on axial CT images (**A**). Valve calcifications (red arrow) were manually selected by the operator and appeared as blue pixels after selection (**B**). Attention was paid not to include aorta calcifications (yellow arrow) in the selection. Multi-planar-Reformation technique (**C**,**D**) was used to improve distinction between aortic (yellow arrow) and valve (red arrow) calcifications.

**Figure 3 jcm-10-04458-f003:**
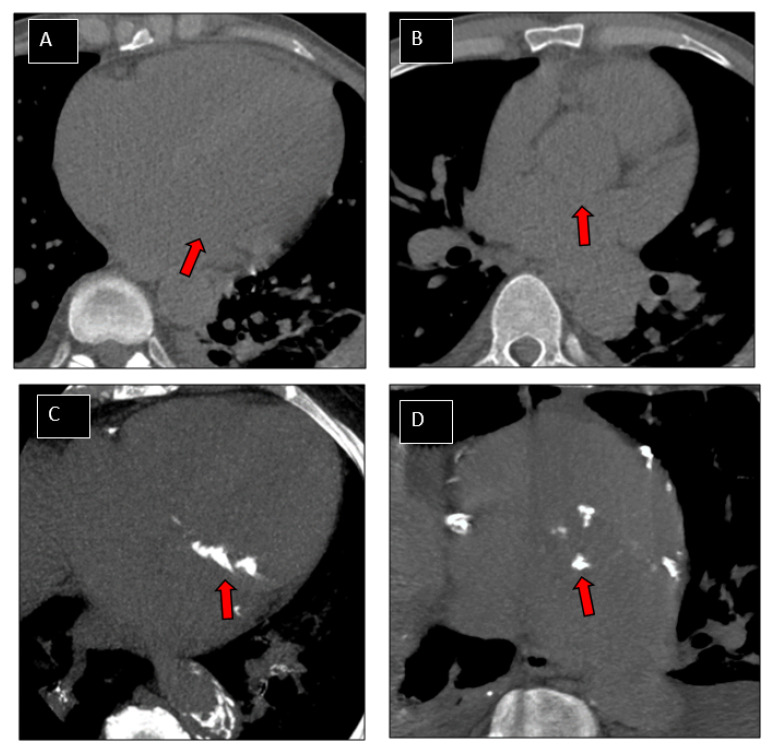
Patterns of patients. Examples of patterns of valve calcifications on CT images. Non-calcified mitral (**A**; arrow) and aortic (**B**; arrow) valves. Valve calcium scores (VCS) were equal to 0 for both valves. Highly calcified mitral (**C**; arrow) and aortic (**D**; arrow) valves. VCS were 1247 and 1080, respectively for mitral and aortic valve.

**Figure 4 jcm-10-04458-f004:**
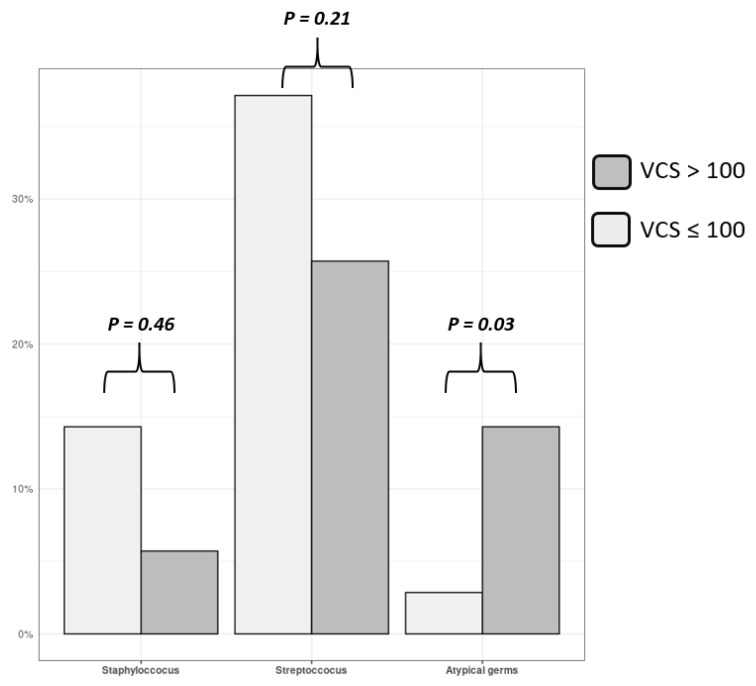
Germ’s type for aortic IE.

**Table 1 jcm-10-04458-t001:** Population characteristics of patients with infective endocarditis (*n* = 70).

		Patients (*n* = 70)
Age, y		75.0 (61.5–83.5)
Gender, Female		28 (40.0)
Bacteria	*Streptoccocus*	34 (48.5)
	*Staphyloccocus*	25 (35.8)
	Other	11 (15.7)
Involved valve	Native aortic valve	36 (51.5)
	Native mitral valve	31 (44.2)
	Native bi-valvular	3 (4.3)
Vegetation size (mm)		13 ± 5
Peripheral embolism		29 (41.4)
Surgery		50 (71.4)
Type of Surgery *	Valve replacement	36 (72.0)
	Plasty	23 (46.0)
Death		14 (20.0)

Data presented as number (%) except for age (median (IQR) and vegetation size (mean ± SD). * Sum of percentages is superior to 100 because of combinate intervention (plasty + replacement) in some patients.

**Table 2 jcm-10-04458-t002:** Characteristics of patients with mitral IE in function of mitral valvular calcium score (VCS).

		Mitral IE (*n* = 34)		*p* value
		High Calcified (VCS > 100) (*n* = 15)	Low Calcified (VCS ≤ 100) (*n* = 19)	
Age, y		80.5 (77.3–85.3)	58.0 (42.5–62.5)	<0.001
Gender, Female		6 (40.0)	7 (36.8)	1
Bacteria	*Streptoccocus*	3 (20.0)	9 (47.4)	0.19
	*Staphyloccocus*	10 (66.6)	7 (36.8)	0.17
	Other	2 (13.3)	3 (15.8)	1
Vegetation (mm)		12 ± 2	14 ± 1	0.55
Peripheral embolism		3 (20.0)	9 (47.4)	0.08
Surgery		6 (40.0)	15 (78.9)	0.03
Type of surgery ^$^	Valve replacement	3 (50.0)	6 (40.0)	1
	Plasty	5 (83.3)	14 (93.3)	
Death		8 (53.3)	2 (10.5)	0.04

Data presented as number (%) except for age (median (IQR) and vegetation size (mean ± SD). VCS = valve calcium score. ^$^ Valve replacement and plasty could be associated, on the same valve or not.

**Table 3 jcm-10-04458-t003:** Characteristics of patients with native aortic IE in function of aortic valvular calcium score (VCS).

		Aortic IE (*n* = 39)		*p* value
		High Calcified (VCS > 100) (*n* = 18)	Low Calcified (VCS ≤ 100) (*n* = 21)	
Age, y		76.5 (69.3–82.0)	54.0 (43–63)	<0.001
Gender, Female		6 (33.3)	9 (42.8)	0.11
Bacteria	*Streptoccocus*	9 (50.0)	14 (66.6)	0.21
	*Staphyloccocus*	3 (16.7)	6 (28.6)	0.46
	Other	6 (33.3)	1 (4.8)	0.03
Vegetation (mm)		13 ± 2	12 ± 1	0.9
Peripheral embolism		5 (27.7)	12 (57.1)	0.13
Surgery		12 (66.6)	17 (80.9)	0.46
Type of surgery ^$^	Valve replacement	12 (100)	15 (88.2)	1
	Plasty	2 (16.7)	2 (11.7)	
Death		3 (16.7)	1 (4.8)	0.32

Data presented as number (%) except for age (median (IQR) and vegetation size (mean ± SD). VCS = valve calcium score. ^$^ Valve replacement and plasty could be associated, on the same valve or not.

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
