# Peer review of "Should We Quantify Valvular Calcifications on Cardiac CT in Patients with Infective Endocarditis?"

_jcm, 2021, doi:10.3390/jcm10194458_

Round 1

Reviewer 1 Report

I have read the manuscript of work "Should we quantify valvular calcifications on cardiac CT in patients with infective endocarditis?"

Authors evaluated the impact of valvular calcifications measured on cardiac computed CCT in patients with IE. Study included 70 patients with native IE Authors showed that the amount of valvular calcifications on CT was associated with embolism risk, rate of surgery and 1-year risk of death in patients with mitral IE, raising the question of their systematic quantification in native IE.

My major remarks:

  • study group was to small
  • single center study
  • not register in Clinical Trial gov.
  • discussion is too short
  • lack of conclusion in main text
  • references are in an unacceptable format

Reviewer 2 Report

  1. Introduction is too long. Cut it down to only 1 paragraph.
  2. Remove sentences 59-67 on page 2.
  3. Page 3, figure 1: Remove the spell check red lines from underlining figure texts.
  4. Did you perform multivariate analysis? If you did, what variables were used for the analysis? Please address that in a table form in the manuscript. Without the multivariate analysis, you cannot conclude that higher VCS had higher embolic and other events. 
  5. Need to add a paragraph on future direction/clinical implication based on the results of the study. How can you help prevent perioperative and postoperative events by measuring valvular calcium score?

Round 2

Reviewer 1 Report

The manuscript has been 
sufficiently improved to warrant publication in JCM